# Hydrothermal Synthesis of 1T-MoS_2_/Pelagic Clay Composite and Its Application in the Catalytic Reduction of 4-Nitrophenol

**DOI:** 10.3390/ma14227020

**Published:** 2021-11-19

**Authors:** Nan Li, Qiwei Sun, Peiping Zhang, Shubo Jing

**Affiliations:** 1Key Laboratory of Automobile Materials of Ministry of Education, School of Material Science and Engineering, Jilin University, 5988 Renmin Street, Changchun 130022, China; lin@jlu.edu.cn (N.L.); sunqw0802@163.com (Q.S.); zhangpp@jlu.edu.cn (P.Z.); 2College of Chemistry, Jilin University, 2699 Qianjin Street, Changchun 130012, China

**Keywords:** molybdenum disulfide, pelagic clay, catalytic reduction, 4-nitrophenol

## Abstract

Pelagic clay is an emerging marine resource with strong hydrophilicity, fine particles and a large specific surface area. In this work, a 1T-MoS_2_/pelagic clay composite was fabricated by hydrothermal synthesis. In the composite, 1T-MoS_2_ nanosheets are evenly dispersed on the surface of the clay minerals, significantly reducing the agglomeration of MoS_2_. Compared with pure 1T-MoS_2_, the 1T-MoS_2_ nanosheets generated on the surface of pelagic clay have significantly smaller lateral dimensions and thicknesses. Moreover, the specific surface area is much larger than that of the pure 1T-MoS_2_ nanosheets fabricated by the same method, indicating that the active sites of the MoS_2_ sheets are fully exposed. In addition, the composite exhibited excellent hydrophilicity, leading to a high dispersibility in aqueous solutions. In this work, the composite was used as a catalyst in the **reduction** of 4-nitrophenol (4-NP), and the conversion of 4-NP reached up to 96.7%. This result shows that the 1T-MoS_2_/pelagic clay composite is a promising catalyst in a variety of reactions.

## 1. Introduction

4-nitrophenol(4-NP) is one of the most prevalent organic pollutants in wastewater resulted from agricultural and industrial sources [1,2,3,4]. Various methods have been applied for the removal of 4-NP pollutants from the environment, mainly including adsorption, microbial degradation, electrochemical treatment and catalytic reduction [5,6,7,8,9]. Direct catalytic reduction of 4-NP to 4-aminophenol (4-AP) is one of the most effective, environmentally friendly and economical methods for 4-NP removal. 4-AP is a very important intermediate for many industrial products, such as paracetamol, painkillers and dyes [10,11]. Compared to the high-temperature and high-pressure gas-phase hydrogenation process with H_2_ as the hydrogen source that involves tedious operations, the liquid-phase reduction process with sodium borohydride (NaBH_4_) as the hydrogen source is more desirable and has been widely investigated due to its mild reaction conditions and simple operation [12,13]. However, 4-NP and NaBH_4_ hardly react in the absence of a catalyst because of unfavorable kinetics. Therefore, an efficient catalyst is necessary for this reaction. The most reported catalysts for the reduction of 4-NP to 4-AP are noble metals such as Au, Ag, Pt and Pd [14,15,16]. Unfortunately, the practical application of these catalysts is limited by the high cost and scarcity. Accordingly, the search for efficient and cost-effective alternative catalysts is essential and critical for environmental protection [17,18,19,20].

As a typical Two-dimensional (2D) layered transition metal disulfide, MoS_2_ is considered to be a promising catalyst for replacing noble metals because its edge sites exhibit catalytic activity similar to platinum [21]. MoS_2_ is usually found in the semiconducting 2H phase (trigonal prismatic coordination) in nature. However, the metallic 1T phase (octahedral coordination) of MoS_2_ (1T-MoS_2_) is of higher electrical conductivity and more catalytic active sites; therefore, it has shown excellent catalytic activity in electrocatalytic reactions [22,23]. In recent years, hydrothermal methods have been developed to fabricate 1T-MoS_2_ except for the traditional exfoliation strategy with lithium ions [24]. However, pure 1T-MoS_2_ nanosheets prepared by hydrothermal methods are prone to agglomerate to form nanospheres, leading to blockage of the active sites and decline in catalytic activity [25,26]. It has been reported that growing 1T-MoS_2_ nanosheets on a substrate like graphene can effectively inhibit the aggregation, improve the dispersion and stability of 1T-MoS_2_ [27,28]. In our previous study, a symbiotic composite of 1T-MoS_2_ and pelagic clay was synthesized by a facile hydrothermal method, and the composite exhibited excellent catalytic activity in the photocatalytic degradation of tetracycline hydrochloride [29].

Pelagic clay is a deep-sea sedimentary clay mineral containing montmorillonite, illite and kaolinite. As an emerging marine resource, pelagic clay has many advantages, such as extremely rich reserves, high hydrophilicity, fine particles, high structural activity and large specific surface area [30].

In this work, a composite of 1T-MoS_2_ with pelagic clays was fabricated by a simple hydrothermal method. 1T-MoS_2_ in the composite accounts for approximately was 57.9%. Moreover, the MoS_2_ nanosheets are relatively uniformly dispersed on the clay minerals surface, which inhibits the agglomeration of the MoS_2_ nanosheets, increases the specific surface area by approximately six times and enhances the exposed catalytic active sites. The composite also exhibits excellent hydrophilicity, which enables a better dispersion in aqueous solution. When it was used in the catalytic reduction of 4-NP, the conversion of 4-NP reached 96.7%.

## 2. Materials and Methods

### 2.1. Materials

Pelagic clay was got from the bottom of the Indian Ocean. Prior to the synthesis, the pelagic clay was thoroughly washed with deionized water, filtered, dried at 80 °C for 24 h and then ground manually with an agate mortar until there was no obvious grainy substance. All of the chemicals were analytical grade reagents. Hexadecyl trimethyl ammonium bromide (CTAB), sodium molybdate (Na_2_MoO_4_·2H_2_O), thiourea ((NH_2_)_2_CS), 4-nitrophenol (4-NP), sodium borohydride (NaBH_4_) and propionic acid were purchased from Chemical Reagent Factory (Beijing, China).

### 2.2. Fabrication of 1T-MoS_2_/PC Composites 

Firstly, 1.2 g pelagic clay was dispersed into the mixture of 0.14 g hexadecyl trimethyl ammonium bromide (CTAB) and 20 mL of deionized water, and then stirred at 60 °C for 24 h. Subsequently, 0.484 g sodium molybdate (Na_2_MoO_4_·2H_2_O), 0.76 g thiourea ((NH_2_)_2_CS) and 16 mL propionic acid were added into the above solution. Finally, the whole mixture was transferred into a 100 mL Teflon-lined stainless-steel autoclave and kept at 180 °C for 4 h. After the reaction, the precipitate was centrifuged and washed with deionized water and ethanol three times and dried in a vacuum oven for 24 h. The product was marked as 1T-MoS_2_/PC.The content of MoS_2_ in the 1T-MoS_2_/PC composite was estimated to be ~20% (wt.%) according to the ratio between the theoretical yield of MoS_2_ and the feeding quality of pelagic clay. For comparison, a pure MoS_2_ sample without pelagic clay was also prepared through a similar process. The product was marked as 1T-MoS_2_.

### 2.3. Material Characterization 

Powder X-ray diffraction (PXRD) (Fangyuan Instrument Co., Ltd., Dandong, China) was tested on a DX2700 diffractometer with a Cu Kα radiation source (λ = 1.5406 nm), with a measurement in 2θ range: 5° to 80°. Scanning electron microscopy (SEM) (Electronics Co., Ltd., Tokyo, Japan) analysis was accomplished on a JEOL JSM-6700F microscope. Transmission electron microscopy (TEM) (Electronics Co., Ltd., Japan) and energy dispersive spectroscopy (EDS) (Electronics Co., Ltd., Japan) characterization were performed on a JEOL JEM-2100F microscope at 200 kV. X-ray photoelectron spectroscopy (XPS) (ThermoFisher, Waltham, MA, USA) analysis was performed on an EscaLab 250Xi electronic spectrometer. Raman spectra were measured using a micro-Raman spectrometer (Renishaw, Wotton-under-Edge, UK) with a laser wavelength of 532 nm at 0.2 mW. A C20001C contact angle meter (Zhongchen Digital Technology Equipment Co., Ltd., Shanghai, China) was used to characterize the surface wettability of samples. UV-vis spectra were recorded in the range of 250–550 nm on a T6-NC (Persee, Beijing, China) spectrophotometer. A JW-BK222 (jingweigaobo, Beijing, China) automated sorption system was used for nitrogen adsorption–desorption measurement. Specific surface area data were obtained by the Brunauer–Emmet–Teller (BET) method. 

### 2.4. Catalytic Activity Evaluation

The reduction of 4-NP to 4-AP was conducted in water in the presence of 1T-MoS_2_/PC using NaBH_4_ as the reductant. The catalytic activity of 1T-MoS_2_/PC was evaluated by measuring the conversion of 4-NP. 

In a typical catalytic reduction reaction, 0.136 g (excess) of sodium borohydride (NaBH_4_) was added to 50 mL of 0.12 mmol/L 4-NP solution and stirred until NaBH_4_ was completely dissolved. Subsequently, 1T-MoS_2_/PC catalyst (with various solid–liquid ratio of 1–6 g/L) was added into the above solution. The mixture was magnetically stirred at various temperatures (20–50 °C) for 15 min. After the reaction, the mixture was separated by filtration. The catalyst powder was recovered, and one milliliter of filtrate was poured into a quartz cell for UV-vis spectroscopy analysis. 

4-NP shows a characteristic peak at 400 nm wavenumber in the UV-vis spectrum, and the intensity of the absorption peak is linearly connected with the concentration of 4-NP. Consequently, the conversion of 4-NP was calculated by measuring the absorbance of the solution at 400 nm before and after the reaction [1,14,31]. The absorptions with various concentrations of 4-NP were measured (from 0.0024 mmol/L to 0.12 mmol/L), and the linear standard equation is fitted with: y = 0.00217 + 17.00795x (where y represents absorbance and x represents concentration of 4-NP in mmol/L). The conversions of 4-NP (C (%)) was calculated according to the following formula: C (%) = ((C_0_ − C_1_)/C_0_) × 100%, where C_0_ and C_1_ is the concentration of 4-NP before and after the reaction, respectively.

## 3. Results and Discussion

### 3.1. Structural Characterization

The crystallographic structures of pelagic clay, pure 1T-MoS_2_ and 1T- MoS_2_/PC were characterized utilizing X-ray diffraction (XRD) measurements. As shown in Figure 1, the XRD pattern of pelagic clay reveals its poor crystallization. Only diffraction peaks from impurities (e.g., quartz, feldspar, kaolinite, calcite) are observed [29]. The XRD pattern of pure 1T-MoS_2_ clearly displays three typical diffraction peaks that correspond to the (002), (100) and (110) planes of MoS_2_, respectively. For 1T-MoS_2_/PC, the characteristic peaks of pelagic clay and two weak diffraction peaks representing the (100) and (110) planes of 1T-MoS_2_ can be observed. However, the peak of (002) crystal plane on the 1T-MoS_2_ is much weaker and almost invisible, indicating that fewer layers of MoS_2_ nanosheets are stacked along the c-axis [1]. The XRD analysis reveals that the presence of pelagic clay reduces the long-range ordering of 1T-MoS_2_ crystals.

To support this hypothesis, we analyzed the microscopic morphology of pure 1T-MoS_2_ and the 1T-MoS_2_/PC composite. The SEM images of 1T-MoS_2_ and 1T-MoS_2_/PC samples are presented in Figure 2. Pure 1T-MoS_2_ exhibits obvious agglomeration of nanosheets, performing a flower-like structure with a diameter of ~500 nm (Figure 2a). The lateral size of each MoS_2_ nanosheet is approximately 200–300 nm. In contrast, in the 1T-MoS_2_/PC composite (Figure 2b), the agglomeration of MoS_2_ nanosheets is almost entirely absent. The MoS_2_ nanosheets in the composite disperse relatively uniformly on the pelagic clay mineral surface. Compared to pure 1T-MoS_2_, the lateral size of the MoS_2_ nanosheets is significantly smaller (~100–200 nm), and their thickness is reduced from ~24 nm to ~11.9 nm, which agrees well with the XRD results. The size reduction of the MoS_2_ nanosheets provides more edges and significantly increases exposed active sites.

TEM and high-resolution transmission electron microscopy (HRTEM) images of the 1T-MoS_2_/PC are shown in Figure 3. The sample has a typical layered structure. The dark areas (Figure 3a) are composed of thick pelagic clay layers, while the flocculent substance with small contrast is a thinner MoS_2_ sheet. A well-stacked layered structure with a lattice spacing of 0.62 nm is evidently visualized in Figure 3b, corresponding to the (002) crystal plane of 1T-MoS_2_. This is solid evidence for the presence of MoS_2_ [1]. Energy dispersive spectroscopy (EDS) confirmed this result. As displayed in Figure 3c–f, the thicker areas of the sample contain mainly Si and Al, which are the major components of the pelagic clay minerals. The surrounding flocculent substance contains mainly S and Mo, suggesting that these areas are mainly MoS_2_. This result also proves the composite structure of MoS_2_ and pelagic clay.

To further investigate the surface elemental composition and the chemical feature, XPS measurements were conducted on 1T-MoS_2_ and 1T-MoS_2_/PC samples under a survey scan that was performed in the range of 0–700 eV. As exhibited in Figure 4a, Al, Si, S and Mo peaks are observed in the XPS spectrum of 1T-MoS_2_/PC. The binding energies of the Mo 3d electrons for both 1T-MoS_2_ and 1T-MoS_2_/PC can be fitted to two pairs of peaks (Figure 4b). The doublet with the higher binding energy is attributed to Mo 3d_5/2_ and 3d_3/2_ orbitals in the 2H phase MoS_2_, indicating that these two samples contain a small amount of the 2H phase [32,33]. This phase is present because the 1T phase is thermodynamically unstable and part of it undergoes a transformation to the more stable 2H phase. These two peaks are located at 229.5 and 232.9 eV for 1T-MoS_2_/PC and are located at 229.8 and 233.2 eV for 1T-MoS_2_, in agreement with the literature data [23]. The pair of peaks with lower binding energies in the spectrum then corresponds to Mo 3d_5/2_ and 3d_3/2_ in the 1T phase MoS_2_. These two peaks lie in at 228.9 and 232.1 eV on 1T-MoS_2_/PC and at 229.1 and 232.3 eV on 1T MoS_2_, respectively. The binding energies were 0.6–0.8 eV lower than that of the corresponding 2H phase MoS_2_, demonstrating that 1T phase MoS_2_ was present in the 1T-MoS_2_/pelagic clay composite. The difference existed in the binding energies was due to the different coordination structures of the 1T and 2H phases [31].

The high-resolution spectra of S 2p (Figure 4c) indicates that the S 2p_3/2_ and S 2p_1/2_ signals of 1T-MoS_2_/PC are located at 161.8 and 163.2 eV, respectively, lower than the binding energies of S 2p_3/2_ and S 2p_1/2_ in 1T MoS_2_ (162.0 and 163.6 eV). Based on the above results, it is concluded that electron transport from the pelagic clay to 1T-MoS_2_ occurred [1,34], proving that the MoS_2_ orbitals are tightly hybridized with those of pelagic clay in the 1T-MoS_2_/PC composite. According to the ratio of the Mo 3d peak area in the XPS spectrum, the 1T phase content in 1T-MoS_2_ was 46.2%, and the 1T phase content in the 1T-MoS_2_/PC composite was 57.9%.

To further verify that the MoS_2_ in the 1T-MoS_2_/PC is found predominantly in the 1T phase, Raman spectroscopy was employed to determine the phase structure of the samples (Figure 5). In addition to the two characteristic peaks located at 385 and 404 cm^−1^ corresponding to the in-plane and interlayer vibrations of 2H-phase MoS_2_, there are two peaks at 235 and 336 cm^−1^, which correspond to the J_2_ and J_3_ vibrational peaks of the 1T-phase MoS_2_, respectively. This result provides strong evidence that 1T-phase MoS_2_ exists in the 1T-MoS_2_/PC composite.

Figure 6 presents the results of the surface wettability. Pure 1T-MoS_2_ exhibits hydrophobicity with a 112.4° water contact angle. In contrast, the contact angle decreased drastically to 34.3° after the sample was hybridized by pelagic clay, suggesting that the pelagic clay significantly enhanced the hydrophilicity of the 1T MoS_2_. This increased hydrophilicity not only improves the dispersion of MoS_2_ in aqueous solutions, but benefits its application as a catalyst in aqueous media.

Additionally, the BET surface areas of the 1T-MoS_2_, 1T-MoS_2_/PC and pelagic clay were measured and were found to be 4.4, 27.0 and 100.4 m^2^ g^−1^, respectively. Based on the analysis of these data, following its hybridization with pelagic clay, 1T-MoS_2_/PC exhibited a 6-fold increase in the BET surface area in contrast with that of pure 1T-MoS_2_, further demonstrating its potential to be an efficient catalyst.

### 3.2. Testing of the Catalytic Behavior of 1T-MoS_2_/PC

We determined the catalytic activity of the samples in reduction reaction in the aqueous phase and with NaBH_4_ as the reductant at 30 °C. The absorbance at 400 nm in the UV-visible spectrum was measured and converted to the concentration of 4-NP. The experimental results suggest that pure pelagic clay was inert in the reduction of 4-NP (Figure 7). Pure 1T-MoS_2_ was catalytically active for the reaction (88.5% conversion of 4-NP). When MoS_2_ was hybridized with pelagic clay, its catalytic reduction activity was significantly enhanced. The conversion of 4-NP was as high as 96.9% within 15 min at 30 °C when the 1T-MoS_2_/PC composite was used as the catalyst.

To further investigate the influence of the mass content of 1T-MoS_2_/PC on the conversion of 4-NP, various quality of 1T-MoS_2_/PC were added to the solution containing 4-NP and NaBH_4_ to reach a solid/liquid ratio of 1, 2, 3, 4, 5 and 6 g/L. The results are shown in Figure 8. The conversion of 4-NP corresponding to the six different content of catalysts is 63.2%, 82.2%, 96.9%, 97.7%, 99.8% and 99.9%, respectively (Figure 9). When the quality of the added catalyst was small, the conversion of 4-NP increased significantly with increasing catalyst quality. After the solid/liquid ratio reached 3 g/L, 4-NP conversions reached more than 95% within 15 min. Then, the catalyst quality continued to increase, and the conversions of the 4-NP increase were not significant. Therefore, a solid/liquid ratio of 3 g/L is the optimal catalyst addition quality in terms of cost.

The effect of the temperature on the conversion of 4-NP was also investigated. The catalytic reduction activity of 1T-MoS_2_/PC was measured at a solid/liquid ratio of 3 g/L for 15 min at 20–50 °C, and the experimental results are shown in Figure 10. The conversion of 4-NP is 88.2%, 96.9%, 98.5% and 99.4% at 20 °C, 30 °C, 40 °C and 50 °C, respectively. The extent of the reaction gradually increased with increasing temperature. At 30 °C, the reaction proceeded up to 96.9%. Further increases in the temperature did not noticeably affect the extent of the reaction. Therefore, 30 °C is a relatively suitable reaction temperature for this reaction.

## 4. Conclusions

In conclusion, a composite catalyst consisting of 1T-MoS_2_ and pelagic clay (1T-MoS_2_/PC) with high catalyst activity was prepared through a simple hydrothermal method. The lateral size and thickness of the MoS_2_ nanosheets in the composite were significantly decreased, and the specific surface area was significantly increased relative to pure MoS_2_ nanosheets. Additionally, the hydrophobicity of MoS_2_ was effectively improved by the excellent hydrophilicity of the pelagic clay itself, enabling it to be better dispersed in solution and enhancing the catalytic activity of the liquid-phase system. The 1T-MoS_2_/PC composite was used for the catalytic reduction of 4-NP to 4-AP. The results revealed that the conversion of 4-NP was as high as 96.9% in 15 min under optimal reaction conditions (30 °C, catalyst solid/liquid ratio of 3 g/L). The research in this paper provides a new method for the enhancement of the catalytic reduction activity of MoS_2_ and suggests opportunities for the high value-adding utilization of pelagic clay.

## Figures and Tables

**Figure 1 materials-14-07020-f001:**
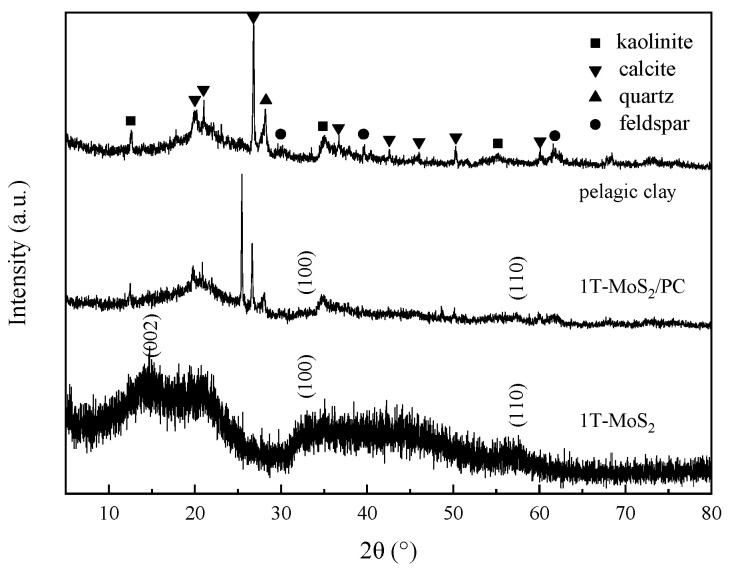
XRD patterns of pelagic clay, 1T-MoS_2_ and 1T-MoS_2_/PC.

**Figure 2 materials-14-07020-f002:**
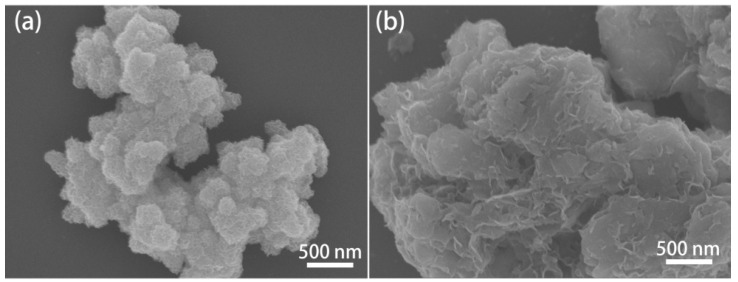
SEM images of (**a**) 1T-MoS_2_ and (**b**) 1T-MoS_2_/PC.

**Figure 3 materials-14-07020-f003:**
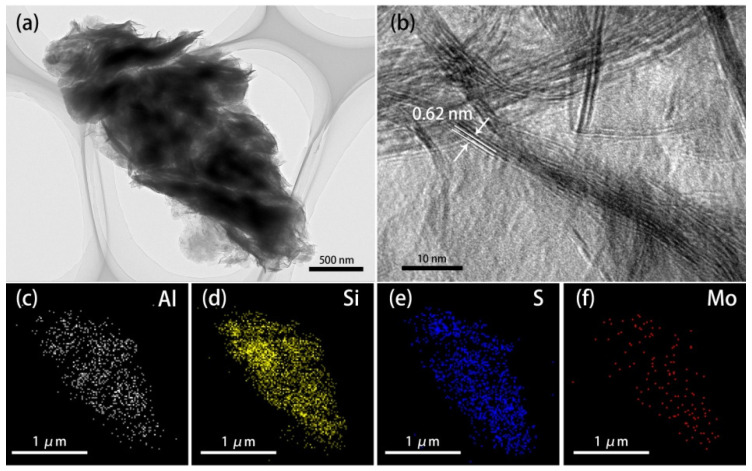
(**a**) TEM image and (**b**) HRTEM image of 1T-MoS_2_/PC, and (**c**–**f**) element distribution of Al, Si, S and Mo in 1T-MoS_2_/PC.

**Figure 4 materials-14-07020-f004:**
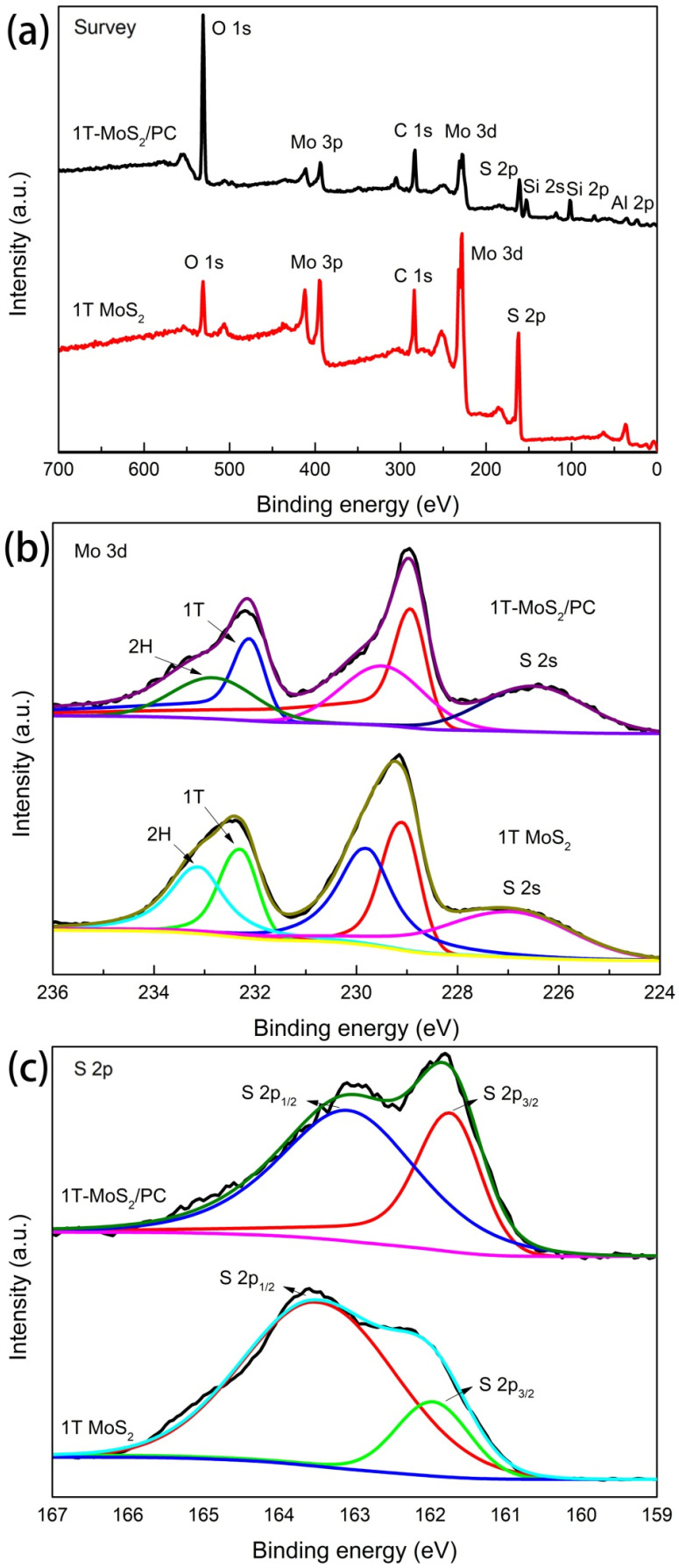
(**a**) XP survey spectra of 1T-MoS_2_/PC and 1T-MoS_2_. High-resolution scans for (**b**) Mo 3d and (**c**) S 2p.

**Figure 5 materials-14-07020-f005:**
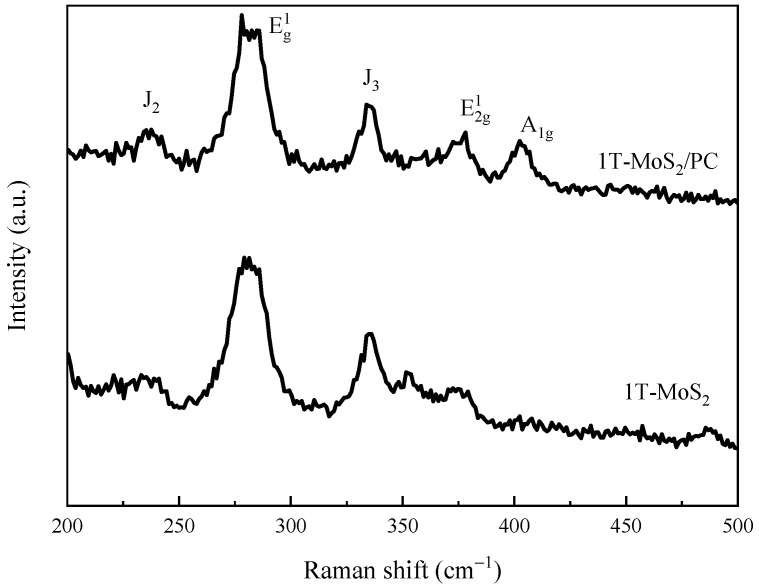
Raman spectra of 1T-MoS_2_/PC and 1T-MoS_2_.

**Figure 6 materials-14-07020-f006:**
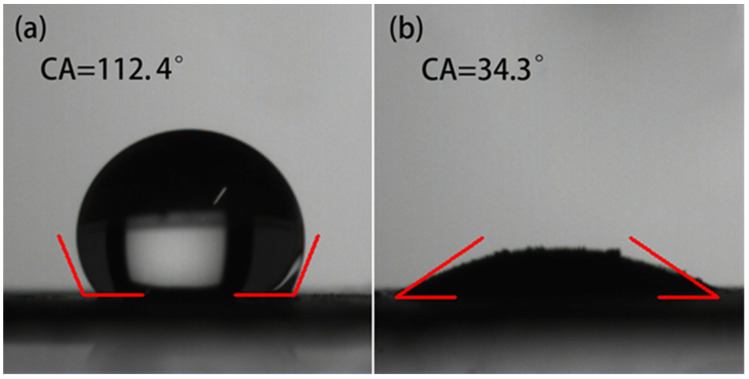
Surface wettability of (**a**) 1T-MoS_2_ and (**b**) 1T-MoS_2_/PC.

**Figure 7 materials-14-07020-f007:**
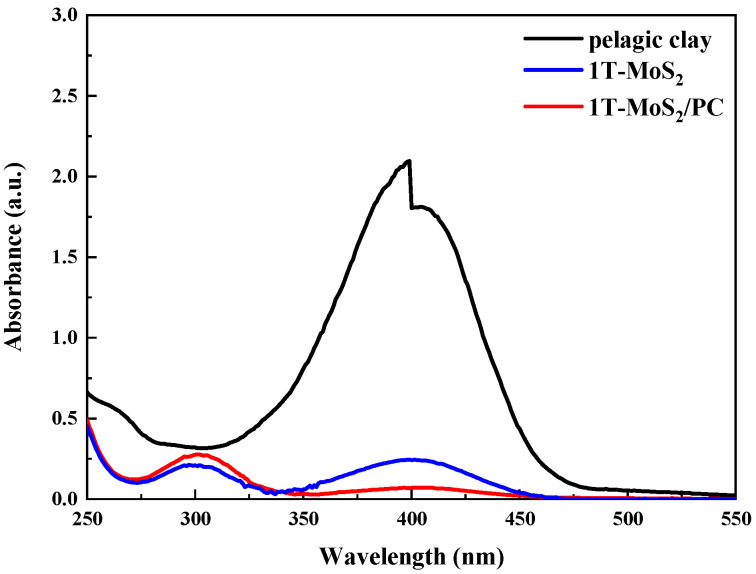
UV–vis absorption spectra of catalytic reduction of 4-NP with NaBH_4_ over pelagic clay, 1T-MoS_2_ and 1T-MoS_2_/PC at a solid/liquid ratio of 3 g/L. (The initial concentration of 4-NP was 0.12 mmol/L).

**Figure 8 materials-14-07020-f008:**
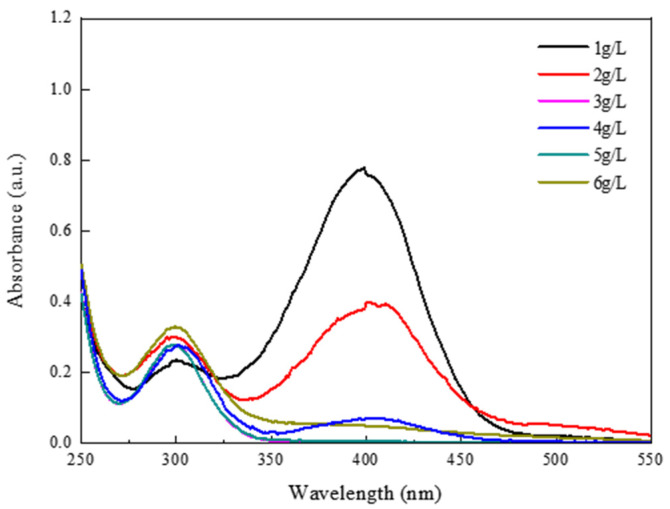
UV–vis absorption spectra of the 4-NP solution with various quality of 1T-MoS_2_/PC. (The initial concentration of 4-NP was 0.12 mmol/L).

**Figure 9 materials-14-07020-f009:**
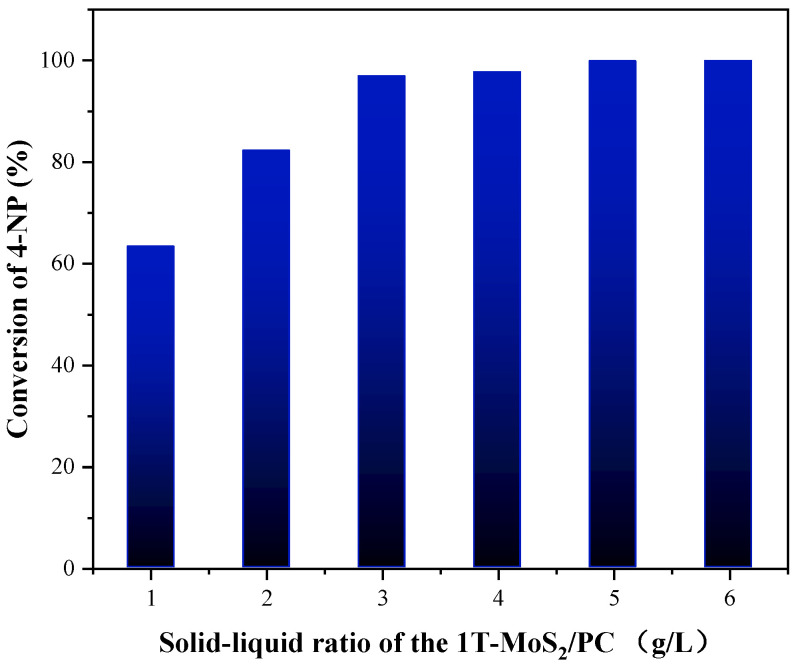
Influence of the 1T-MoS_2_/PC content on the catalytic reduction of 4-NP.

**Figure 10 materials-14-07020-f010:**
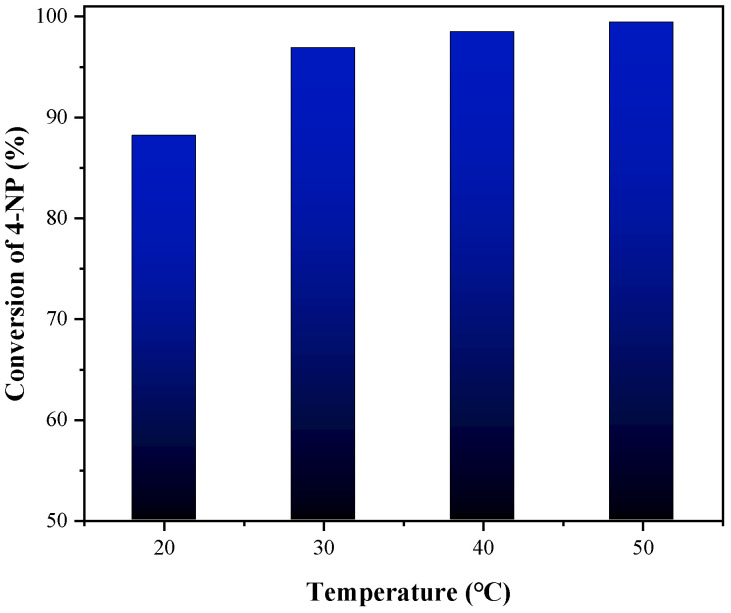
Effect of temperature on catalytic reaction of 1T-MoS_2_/PC.

## Data Availability

The data presented in this study are available on request from the corresponding author.

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
