# Peer review of "Hydrothermal Synthesis of 1T-MoS2/Pelagic Clay Composite and Its Application in the Catalytic Reduction of 4-Nitrophenol"

_materials, 2021, doi:10.3390/ma14227020_

Round 1

Reviewer 1 Report

The manuscript “Hydrothermal synthesis of 1T-MoS2/pelagic clay composite and its application in the catalytic reduction of 4-nitrophenol” by N.Li, Q.Sun, P.Zhang, and S.Jing is an interesting work, which has great importance for the improvement of the ecological situation, in particular the removal of 4-nitrophenol from water. The advantage of this work is the use of the emerging marine resource – pelagic clay - as a support. The highly efficient catalyst comprising MoS2 incorporated/deposited onto pelagic clay, as well as the hydrothermal method of its preparation, are proposed.

The manuscript can be accepted to the publication in the Materials after minor revision (please see the remarks below).

The text of the manuscript contains some incorrect sentences and the phrases, which are marked by yellow highlighter in the attached pdf-file, for example:
- Lines 57-58: “composite composed” 
- Lines 86-87: “mass ratio of MoS2/pelagic clay of 20% with content of MoS2”
- Lines 150-151: “The sample shows typical layered structure” 
- Line 175: “small amount of conversion”
- Line 228: “converted to the conversion”

Also, I would like to make some stress on the following:
1. The content of MoS2 used in the T-MoS2/PC composite is not explained in the text of the manuscript.
2. Line 101: Please, check the power of the laser.
3. The content of Figure 7 and its caption do not correspond to each other.
4. Line 252: “Influence of the 1T-MoS2/PC concentration”, the caption to figure 9 is not complete.

Reviewer 2 Report

In this article, a compound of MoS2 and clay was characterized by one-step hydrothermal synthesis. The composite material was used as a catalyst in the reduction of 4-nitrophenol. The characterization of the composite is complete and well structured. However, the study of the reduction of 4-nitrophenol is brief, a statistical treatment of the results would be convenient. This corroborates the catalytic behavior of the compound.

There are some points that, in my opinion, need to be clarified in detail:
In the section:

  • 2.4. Evaluation of catalytic activity.
    • A more detailed description of the 4N reduction conditions is necessary. For example: pH, agitation, volume ...
  • 3.2. Catalytic behavior test 1T-MoS2 / PC
    • "Figure 7 and 8.
      • The UVVis spectrum is necessary to obtain between 250 and 550nm so that the jump of the lamp change does not come out. Indicate the concentration of the species.
    • Figure 9 and 10.
      • It is necessary to indicate the standard deviation of the results.

Reviewer 3 Report

General Comment:

The research aims at the catalytic reduction of 4-nitrophenol utilizing Molybdenum disulfide (MoS2) composite with pelagic clays mineral. First, analytical techniques (SEM and XRF) were used to characterize the prepared composites. Later, the catalytic activity of the composite material was examined to reduce 4-nitrophenol, and the effects of several parameters (composite concentration and temperature) on the reduction were determined. The results are reasonable, and the topic is intriguing, but specific crucial points or insights might be highlighted for more clarity. The manuscript is considered publishable after the following significant revisions.

Comments:

  1. Page 2, Line 76, Is it necessary to pre-treat pelagic clay prior to evaluating its catalytic activity?
  2. Page 3, Section 2.4, i.e., assessment of catalytic activity, is ambiguous. For instance, was there any mixing occurring throughout the activity evaluation process? What method did the authors use to separate the composite catalyst? How were samples collected to determine the strength of the 4-NP absorption peak?
  3. Did the authors use any analytical procedures to validate the conversion of 4-NP to 4-AP?
  4. Is it feasible to incorporate the reusability of the composite catalyst for catalytic reduction of 4-nitrophenol?
  5. Figure 8, There is no absorbance scale.
